# SARS-CoV-2 mRNA Vaccine-Induced Cellular and Humoral Immunity in Hemodialysis Patients

**DOI:** 10.3390/biomedicines10030636

**Published:** 2022-03-09

**Authors:** Ewa Kwiatkowska, Krzysztof Safranow, Iwona Wojciechowska-Koszko, Paulina Roszkowska, Violetta Dziedziejko, Marek Myślak, Jacek Różański, Kazimierz Ciechanowski, Tomasz Stompór, Jarosław Przybyciński, Piotr Wiśniewski, Norbert Kwella, Sebastian Kwiatkowski, Tomasz Prystacki, Wojciech Marcinkowski, Leszek Domański

**Affiliations:** 1Clinical Department of Nephrology, Transplantology and Internal Medicine, Pomeranian Medical University, 70-111 Szczecin, Poland; jacekrozanski@wp.pl (J.R.); kazcie@pum.edu.pl (K.C.); jarpe85@gmail.com (J.P.); piotr_wi@yahoo.pl (P.W.); domanle@pum.edu.pl (L.D.); 2Department of Biochemistry and Medical Chemistry, Pomeranian Medical University, 70-111 Szczecin, Poland; chrissaf@mp.pl (K.S.); viola@pum.edu.pl (V.D.); 3Independent Laboratory of Immunological Diagnostics, Department of Microbiology, Immunology and Laboratory Medicine, Pomeranian Medical University, 70-111 Szczecin, Poland; iwonakoszko@interia.pl (I.W.-K.); ppaulinaroszkowska@gmail.com (P.R.); 4Department of Nephrology and Kidney Transplantation, Provincial Integrated Hospital, Arkońska 4, 71-455 Szczecin, Poland; myslak@spwsz.szczecin.pl; 5Department of Nephrology, Transplantology and Internal Diseases, University of Warmia and Mazury, 10-719 Olsztyn, Poland; stompin@mp.pl (T.S.); norbert.kwella@gmail.com (N.K.); 6Department of Obstetrics and Gynecology, Pomeranian Medical University, 70-111 Szczecin, Poland; sebastian.kwiatkowski@pum.edu.pl; 7Fresenius Nephrocare, 60-118 Poznań, Poland; tomasz.prystacki@fmc-ag.com (T.P.); wojciech.marcinkowski@fmc-ag.com (W.M.)

**Keywords:** SARS-CoV-2, mRNA vaccination, humoral immunity, cellular immunity

## Abstract

Background/Aims: Chronic kidney disease CKD patients on intermittent hemodialysis IHD are exposed to SARS-CoV-2 infection and carry a risk of developing severe symptoms. The aim of this study was to evaluate the humoral and cellular immunity induced by two doses of mRNA vaccines, the Pfizer-BioNTech (Comirnaty) COVID-19 Vaccine and the Moderna (mRNA-1273) COVID-19 vaccine. Patients and methods: The study included 281 patients from five dialysis centers in northern Poland. Within 2 weeks prior to the first dose of the vaccine, a blood sample was collected for an evaluation of SARS-CoV-2 antibodies. Thirty to forty-five days after the second dose of the vaccine, a blood sample was taken to evaluate humoral and cellular response. Results: Patients with stage 5 CKD on IHD were characterized by a considerable SARS-CoV-2 vaccine-induced seroconversion rate. The strongest factors influencing the antibodies AB level after vaccination were a pre-vaccination history of SARS-CoV-2 infection, age, the neutrophil-to-lymphocyte ratio NLR, neutrophil absolute count, and the hemoglobin level. Cellular immunity was higher in patients with a pre-vaccination history of SARS-CoV-2 infection. Cellular immunity depended on the albumin level. Positive cellular response to vaccination was a positive factor reducing all-cause mortality, except for COVID-19 mortality (no such deaths were reported during our follow-up). Cellular immunity and humoral immunity were positively mutually dependent. High levels of albumin and hemoglobin, low neutrophil count, and a reduced NLR, translated into better response to vaccination. Conclusions: Patients with stage 5 CKD on IHD were characterized by a considerable SARS-CoV-2 vaccine-induced seroconversion rate and a good rate of cellular immunity. The factors that change with exacerbating inflammation and malnutrition (albumin, hemoglobin, neutrophil count, the NLR) affected the efficacy of the vaccination.

## 1. Introduction

The SARS-CoV-2 coronavirus belongs to the coronavirus (Coronavridae) family first identified in 1960s [1,2,3]. Until recently, members of the family were responsible for typically mild respiratory infections in humans. The 21st century has seen the emergence of new human coronavirus species that most probably developed as a result of mutations of animal viruses. In 2020, the first cases of a SARS-CoV infection causing severe acute respiratory syndrome (SARS) were reported. The pathogen was later given the official name of SARS-CoV-2 (SARS coronavirus 2). It has been responsible for the ongoing epidemic of the COVID-19 (coronavirus disease 2019) infectious disease. According to the WHO data, there have been more than 250 million SARS-CoV-2 infections and more than 5 million related deaths worldwide to date, i.e., 11 November 2021 (https://covid19.who.int/, accessed on 30 January 2022).

Patients with chronic kidney disease (CKD) on intermittent hemodialysis (IHD) face a high risk of contracting SARS-CoV-2 infection. This type of renal replacement therapy entails collective transportation of patients to and from a dialysis center three times a week. In addition, the patients are in contact with other patients and medical staff during the procedure itself, which lasts 3–4 h. IHD patients are unable to stick to the self-isolation rules that are recommended to people with diseases and the elderly. For example, data from France and New York suggest that IHD patients were diagnosed with COVID-19 5 to 16 times more often than the general population at the same time and in the same place. Anti-SARS-CoV-2 antibody tests show that the proportion of CKD patients on IHD who are COVID-19 convalescents is significantly higher, ranging from 28% to 36% (UK and US data) [4,5].

Patients undergoing IHD are not only more exposed to SARS-CoV-2 infection but also carry a high risk of developing severe symptoms. Such patients, when hospitalized for severe symptoms of the disease, are between 30 and 130% more likely to die when compared to patients hospitalized due to COVID-19 but having no comorbid stage 5 CKD. In this group, the SARS-CoV-2 mortality rate varies from 16 to 32%. The increased mortality of SARS-CoV-2 in IHD patients is associated with a weakened immune system, but also with a higher incidence of co-morbidities recognized as risk factors for developing severe symptoms. These include arterial hypertension, obesity, diabetes, and cardiovascular diseases. One out of 33 hemodialysis patients died in the UK during the first wave of the epidemic. Young patients are characterized by extremely high mortality rates in COVID-19 compared to their healthy age-matched peers. The mortality rate in older IHD patients is higher than in their healthy age-matched peers, but the difference is not as significant as in the case of young people. The relative risk of death due to SARS-CoV-2 infection compared to the age-matched general population is 432 for the 20- to 39-year-olds, 94 for the 40- to 59-year-olds, 33 for the 60- to 79-year-olds, and 10 for persons aged 80 or more. This means that young stage 5 CKD patients on IHD are 432 times as likely to die as their age-matched peers [6,7,8]. Such a high risk of death in COVID-19 in a group that meets three times a week is a cause of great anxiety. For this reason, these patients are likely to skip their hemodialysis (HD) sessions, which only increases mortality in this group. In addition, the need to isolate positive patients often makes it impossible for them to undergo the three 4-h-long sessions. The arguments cited above convinced the Polish Ministry of Health that there was a need to promptly vaccinate the whole group of these patients—and not merely those aged 70 or more—against the coronavirus. In Poland, CKD patients on IHD were found to be highly exposed to both contracting the infection and developing its severe symptoms, resulting in the introduction of widespread vaccination in the patients’ own dialysis stations administered by their regular medical staff.

Our knowledge of the effects of vaccinating stage 5 CKD patients on HD is mainly based on the widespread HBV vaccination campaign. Such vaccination is necessary as the patients are exposed to contact with blood through the dialysis equipment and through the frequent use of needles. The currently applicable vaccination schedules have a 95% efficacy against HBV in healthy people. The seroconversion rate in stage 5 CKD patients vaccinated with the standard doses on a standard schedule (20 µg doses of recombinant HBV surface antigen on a 0-, 1-, and 6-month schedule) was unsatisfactory. Therefore, a new strategy for this group of patients has been introduced with doses of 40 µg of recombinant HBV surface antigen on a 0-, 1-, 2-, and 6-month schedule. Ten international units per milliliter has been considered to be the antibody (AB) level that protects against the infection. The available literature implicates that despite the introduction of a new vaccination schedule against HBV, the outcomes remain unsatisfactory. A large analysis of the 2020 databases involving 6628 stage 5 CKD patients on HD concluded that seroconversion occurred in 69% of them. After considering a number of different factors, younger patients without comorbid diabetes, with better hemodialysis adequacy measured using the Kt/V parameter, higher albumin and hemoglobin concentrations, and a lower calculated risk of death, were found to respond better to the vaccination. The outcome of this study was later confirmed by other researchers finding that age is the strongest factor in the development of vaccine-induced seroconversion [9]. Cordova concluded that seroconversion depends on patient age, albumin concentration, urea level prior to HD, co-morbidities, vascular access (a fistula vs. a dialysis catheter in favor of a fistula), and the type of hemodialysis [10].

As shown above, stage 5 CKD patients on IHD are exposed to SARS-CoV-2 infection and carry a risk of developing severe symptoms. Immune system disorders affecting innate and adaptive immunity are known to be significant in this group and to be caused by uremic toxins, the hemodialysis procedure itself, and co-morbidities. The question to be raised is whether the standard manufacturer-recommended dose of the SARS-CoV-2 vaccine will be sufficient to ensure a significant proportion of patients develop seroconversion protecting them from the infection. The aim of this study is to evaluate the humoral and cellular immunity induced by two doses of mRNA vaccines, the Pfizer-BioNTech (Comirnaty) COVID-19 Vaccine, Pfizer Inc. 235 East 42nd Street New York, NY 10017, USA and the Moderna (mRNA-1273) COVID-19 vaccine, One Upland Rd, Norwood, MA 02062, USA.

## 2. Materials and Methods

### 2.1. Patients

The study included 281 patients from 5 dialysis centers in northern Poland (Szczecin Poland, Drawsko Pomorskie Poland, Olsztyn Poland). The patients involved in the study were treated in line with the Declaration of Helsinki and the Declaration of Istanbul. The local ethics committee of the Pomeranian Medical University, Szczecin, Poland, approved the study protocol–KB-0012/06/2021, 26APR2021. The inclusion and exclusion criteria of the patients in the study were as follows:

Inclusion criteria:Obtaining written consent for study enrollment;Age > 18 years;Consent to vaccination;Vaccination with one of the available mRNA vaccines, with two doses administered at least 21 days apart;HD treatment for at least 1 month.

Exclusion criteria:Refusal to participate in the study;Incomplete vaccination—less than two doses;Interval between two doses of vaccination greater than 60 days.

Within 2 weeks prior to the first dose of the vaccine, blood samples were collected for an evaluation of SARS-CoV-2 antibodies (the presence of which would have been a sign of past SARS-CoV-2 infection). Each patient included in the study was inoculated with one of the available mRNA vaccines, with two doses administered at least 21 days apart. Thirty to forty-five days after the second dose, a blood sample was taken to evaluate humoral and cellular response. Factors that could potentially affect seroconversion were assessed. Such factors taken into account were sex, age, the BMI, time since the first hemodialysis, co-morbidities (cardiovascular diseases, diabetes, arterial hypertension, autoimmune diseases), nutritional status, the hemodialysis schedule (time, frequency), and vascular access (arteriovenous fistula vs. dialysis catheter). Additional tests were performed to evaluate blood count (lymphocyte, neutrophil, and HGB levels), the levels of albumin, parathormone, ferritin, transferrin, urea, and creatinine before HD, as well as the HD adequacy measure of Kt/V. An earlier response to HBV vaccination was also assessed to see whether the patient had formed antibodies following the standard vaccination schedule for hemodialysis patients and to identify their antibody titer achieved (patients with a past HBV infection (anti-HBC IgG positive patients) did not have their anti-HBs antibody titer determined). The past infection was assessed for its impact on the immune system response to the vaccination. The study group’s characteristics are shown in Table 1.

### 2.2. Methods

#### 2.2.1. Humoral Response

The humoral response was assessed qualitatively as the presence of IgG seroconversion (present/absent seroconversion) and quantitatively by measuring the concentration of anti-SARS-CoV-2 S protein IgG antibodies. Four-milliliter native blood samples were collected. After centrifugation, anti-SARS-CoV-2 IgG antibodies in the serum were determined using chemiluminescence technology. The LIAISON SARS-CoV2 Trimerics IgG (REF 311520) assay by DiaSorin Inc. (Northwestern Ave–Stillwater, MN, USA) was used for the evaluations. The lower quantification range was 3.8 AU/mL. The upper range was unlimited. In the event of a high titer, the sample was diluted according to the manufacturer’s recommendation.

#### 2.2.2. Cellular Response

Four-milliliter whole-blood samples were collected into lithium heparin tubes. These tubes were then stored at room temperature for up to 12 h, with the blood incubated in three tubes: one basic tube with no antigen, one mitogen-stimulated (control) tube, and one SARS-CoV-2 S protein tube. After a 24-h incubation, the tubes were centrifuged, and interferon-gamma (IFNγ) was determined in the resultant plasma. The cellular response was assessed as the T-cell response to the SARS-CoV-2 antigen (S protein) using IGRA—an interferon-gamma release assay that assesses the concentration of interferon released in response to the viral antigen. The assay assesses basic activity—the IFNγ level without antigenic stimulation. The interferon concentration in this sample had to be ≤400 mIU/mL (if it was higher, the sample was rejected and another one was collected). The second sample was SARS-CoV-2 antigen-stimulated. The IFNγ concentration in the primary sample was subtracted from the concentration obtained for this sample. The result thus obtained was then analyzed. According to the manufacturer’s recommendation, concentrations < 100 mIU/mL were considered as negative, values between 100–200 mIU/mL as borderline, and levels > 200 mIU/mL as positive. For us to be able to assess whether the blood sample contained a sufficient number of immune cells, one of the tubes included the stimulant mitogen. The IFNγ concentration in this sample had to be ≥400 mIU/mL (if it was lower, the sample was rejected and another one was collected).

#### 2.2.3. Statistical Analysis

We used Statistica software (StatSoft, Tulsa, OK, USA) for statistical analysis. The Shapiro–Wilk test was used to study the distribution. The distribution of the evaluated data was significantly different from normal (*p* < 0.05). We used a nonparametric Mann–Whitney U test to compare the two groups. Correlations were studied by means of Spearman’s rank correlation test. Data that were not normally distributed were shown as the median [minimum–maximum]. *p*-values were significant if they were <0.05.

## 3. Results

The study included 281 HD patients from five dialysis centers in Poland: 72 from the first dialysis center, 42 from the second dialysis center, 48 from the third dialysis center, 66 from the fourth dialysis center, and 53 from the fifth dialysis center. There were 112 women and 169 men in the group. Two hundred and sixty-nine of them received the Comirnaty vaccine from Pfizer-BioNTech, and 12 were given the Moderna jab. This group was tested for their anti-S protein antibody concentrations within 2 weeks prior to vaccination to determine if they were COVID-19 convalescents. One hundred and thirteen patients were found to have anti-SARS-CoV-2 S protein antibodies and 168 patients were found to have none. This means that 40.2% of the study subjects had a past SARS-CoV-2 infection. Only 83 patients had a positive history of the infection, representing 73.5% of the patients with anti-SARS-CoV-2 antibodies prior to vaccination. The causes of renal failure were divided into autoimmune, metabolic, and other causes. The first group included 24, the second 142, and the third 115 patients. They were dialyzed using the dialysis catheter (115 people) or the arteriovenous fistula (166 people). Nine patients were on immunosuppressive drugs due to co-morbidities or a renal graft. During the 6-month follow-up, 8 patients contracted COVID-19. During the 6-month follow-up, 12 patients died for reasons not attributable to COVID-19.

Only two patients out of the 281 failed to form antibodies after vaccination—having a level < 3.8 AU/mL. Cellular resistance was determined for 202 patients, with negative results in 38, borderline results in 18, and positive values in 146. This means that unsatisfactory cellular response was found in 28% of our patients. The AB levels did not vary between women and men. The AB levels at 1 month after the second dose were higher in the Moderna (mRNA-1273) group than in the Comirnaty (BNT162b2) group. Persons who had a past SARS-CoV-2 infection prior to vaccination (with present IgG seroconversion) had higher antibody levels and higher IFNγ concentrations at 1 month after the second dose of their vaccine (Figure 1).

As recommended by the manufacturer of the SARS-CoV-2 cellular immunity determination kit, IFNγ concentrations >200 mIU/mL were considered as positive, those between 100–200 mIU/mL as borderline, and those <100 mIU/mL as negative. According to these ranges, the study group was divided into three sub-groups. No differences in anti-SARS-CoV-2 antibody concentrations were found between the negative and the borderline groups. Differences in humoral immunity were found between the negative and the positive groups: the positive group had significantly higher concentrations of antibodies before and after vaccination. Similarly, differences were observed between the borderline and the positive groups, with the positive one having higher antibody concentrations before and at 1 month after the second dose of the vaccine (Table 2, Figure 2).

No differences were found in antibody concentrations before and after vaccination or IFNγ concentrations after vaccination depending on the cause of renal disease, the presence of a dialysis catheter, co-morbidities, or the immunosuppressive drugs administered. During the 6-month follow-up, 12 patients died for reasons not attributable to COVID-19. Those who died within 6-months of being vaccinated had lower cellular immunity at 1 month after the second dose. In addition, they had lower albumin levels, shorter weekly times on HD, higher neutrophil levels, and lower creatinine contents prior to hemodialysis (Table 3).

During the 6-month follow-up, eight people contracted COVID-19. None of them had severe symptoms and there were no deaths due to the infection. The ill patients did not vary in AB levels before and after vaccination, nor cellular resistance levels after vaccination. Their group varied with regard to albumin and HGB levels (Table 4).

The study group was divided into terciles depending on the antibody concentrations at 1 month after the 2nd dose of the vaccine: group 1 (<237 AU/mL), group 2 (237–970 AU/mL), and group 3 (>970 AU/mL). Between the extreme groups (group 1 versus 3), group 3 was found to have higher anti-SARS-CoV-2 antibody concentrations prior to vaccination, higher cellular response, higher creatinine levels before HD, longer times on HD, lower neutrophil concentrations, and larger differences between their pre- and post-vaccination AB levels (Figure 3, Figure 4, Figure 5, Figure 6 and Figure 7). Between groups 1 and 3, and between groups 2 and 3, differences were observed in AB levels before vaccination, cellular immunity, and the delta of AB levels prior to and after vaccination.

Since a history of SARS-CoV-2 infection prior to vaccination was a factor affecting cellular and humoral immunity after vaccination, patients who had a negative anti-SARS-CoV-2 antibody result prior to vaccination were analyzed (*n* = 237). The group was divided into two subgroups depending on the antibody median: group 1 with an AB < 570 AU/mL and group 2 with an AB ≥ 570 AU/mL. Significant differences were found between these groups with regard to the degree of cellular immunity and age (Table 5).

This group was divided into terciles depending on the antibody concentrations at 1 month after the 2nd dose of the vaccine: group 1 (<237 AU/mL), group 2 (237–970 AU/mL), and group 3 (>970 AU/mL). Between the extreme groups (group 1 versus 3), group 3 was found to have a higher cellular response, lower neutrophil concentrations, and lower neutrophil-to-lymphocyte (NLR) ratio values (Figure 8, Figure 9 and Figure 10).

### 3.1. Correlations

#### 3.1.1. The Entire Group

A negative correlation was found between age and anti-SARS-CoV-2 antibody levels at 1 month after completion of the vaccination course (*p* = 0.04; R −12) (Figure 11). The correlation is statistically significant, although the strength of the relationship between the examined factors was weak.

A negative correlation was found between age and AB levels before and after completion of the vaccination course (*p* = 0.026; R −0.14). The correlation is statistically significant, although the strength of the relationship between the examined factors was weak (Figure 12).

A positive correlation was found between pre-vaccination AB levels and the AB levels after completion of the vaccination course (*p* = 1.1 × 10^–^^14^), R 0.47 (Figure 13).

Pre-vaccination AB levels correlated positively with the post-vaccination cellular response rate (*p* = 7.95 × 10^–^^09^, R 0.43) (Figure 14).

A positive correlation was found between AB levels at 1 month after vaccination and the cellular response rate (*p* = 8.6 × 10^–^^11^, R 0.44) (Figure 15).

A positive correlation was found between post-vaccination AB levels and the weekly hours on HD (*p* = 0.017, R 0.14) (Figure 16). The correlation is statistically significant, although the strength of the relationship between the examined factors was weak.

A negative correlation was found between post-vaccination AB levels and absolute neutrophil count (*p* = 0.03, R −0.13). The correlation is statistically significant, although the strength of the relationship between the examined factors was weak (Figure 17).

A positive correlation was found between post-vaccination AB levels and creatinine concentrations (*p* = 0.018, R 0.17). The correlation is statistically significant, although the strength of the relationship between the examined factors was weak (Figure 18).

Cellular immunity, expressed as the IFNγ level, correlated positively with albumin concentrations (*p* = 0.02, R 0.16). The correlation is statistically significant, although the strength of the relationship between the examined factors was weak (Figure 19).

#### 3.1.2. The Group of Patients with No History of SARS-CoV-2 Infection Prior to Vaccination

A negative correlation was found between age and anti-SARS-CoV-2 AB levels at 1 month after completion of the vaccination course (*p* = 0.006, R −0.24). A positive correlation was found between cellular and humoral immunity against SARS-CoV-2 at 1 month after completion of the vaccination course (*p* = 0.02, R 0.23). A positive correlation was found between anti-SARS-CoV-2 antibody levels at 1 month after completion of the vaccination course and the weekly time on HD (*p* = 0.02, R 0.2). A negative correlation was found between anti-SARS-CoV-2 antibody levels at 1 month after completion of the vaccination course and the absolute neutrophil count (*p* = 0.003, R −0.25).

A negative correlation was found between anti-SARS-CoV-2 antibody levels at 1 month after completion of the vaccination course and the NLR (*p* = 0.02, R −0.20). A positive correlation was found between anti-SARS-CoV-2 antibody levels at 1 month after completion of the vaccination course and HGB concentrations (*p* = 0.042, R 0.21).

## 4. Discussion

### 4.1. Humoral Immunity

As mentioned in the introduction, CKD patients on IHD were diagnosed with COVID-19 5 to 16 times more often than the general population at the same time and in the same region (data from France and New York). Anti-SARS-CoV-2 antibody tests show that the proportion of CKD patients on IHD who are COVID-19 convalescents is significantly higher, ranging from 28% to 36% (UK and US data) [4,5]. In our study group, the proportion of patients with a pre-vaccination history of SARS-CoV-2 infection was 40.2%. This is a significant share, which may result from the fact that Poland has already experienced three waves of infections (over a period of more than a year). Seventy-three percent of this group developed symptoms and had their infections confirmed via PCR. Only two patients out of the entire study group failed to develop antibodies. The percentage of seroconversion in stage 5 CKD patients on IHD was 99.3%. In the group studied by Tylicki et al. (patients dialyzed in an HD center in Poland), the percentage of persons with a present seroconversion was 95.6%. In a group of HD patients in Israel, this percentage was 94% [11]. In a group of HD patients in Germany (81 people), seroconversion occurred in 95.1% of them [12]. The very high seroconversion rate (nearly 100%) observed in our study may have been linked to the high share (40.2%) of persons with a pre-vaccination history of infection. In the study by Clement Dantau (France), only 81% of the patients showed positive reconversion. In his study, Melin considered an IgG antibody level <50 AU/mL as negative, a level between 50–100 AU/mL as borderline, and only a level >100 AU/mL as positive. If assessed in the same manner, our study group had 95.3% of patients with a positive, 8.9% with a borderline, and 4.3% with a negative response. In the Melin group, only 74% of patients developed a positive response with AB levels above 100 AU/mL. However, only 5 patients (10%) in his study group had a pre-vaccination history of the infection [13]. In the group studied by Wilde et al. (California), of 610 patients who had received two doses of the vaccine, 14.5% did not demonstrate seroconversion, or their response was considered insufficient (one weakness of the study was that it involved a semi-quantitative assessment) [14]. In the present study, our comparison of patients who had a pre-vaccination history of COVID-19 with those who did not show the former to have significantly higher AB levels after vaccination. This result was also confirmed by the very strong positive correlation between pre-vaccination AB levels and the AB levels after completion of the vaccination course (*p* = 1.1 × 10^−14^; R 0.47) (Figure 1 and Figure 13). Similarly, a pre-vaccination infection increased the cellular response rate after vaccination (*p* = 7.95 × 10^−09^; R 0.43). In the research carried out by Nacasch in Israel, their 105-strong HD group included patients who had previously had COVID-19 and who had higher AB levels, as well [11]. Similar results were obtained by Attias and Broseta [14,15,16]. In our group of 169 patients who had no history of SARS-CoV-2 infection, two of them failed to develop seroconversion. Patients with an antibody titer <50 AU/mL accounted for 11.8%, those with a titer between 50 and 100 AU/mL (borderline response) accounted for 7.87%, and those with a titer >100 AU/mL accounted for 88.2%. These findings are more consistent with those obtained by other researchers, although still, a very large proportion of our patients were able to develop sufficiently high AB levels. The median and SD post-vaccination AB levels across the group were 576.9 and 2639.68 AU/mL, respectively. Depending on the AB level, the group was divided into terciles to form a group with an AB titer <237 AU/mL (94 patients), a group with an AB titer of 237–970 AU/mL (93 patients), and a group with an AB titer > 970 AU/mL (94 persons). The extreme groups were significantly different with regard to the pre-vaccination AB levels and the cellular response rates assessed by the IFNγ level following viral antigen induction, the weekly time on HD, neutrophil count, and creatinine concentrations prior to HD (Figure 3, Figure 4, Figure 5, Figure 6 and Figure 7). The longer the weekly time on HD, the better the humoral response was to vaccination was achieved. Rather than dialysis quality—as no link was identified between the Kt/V parameter and the post-vaccination AB level—this outcome can be explained by the fact (as evidenced by the negative correlation between the weekly time on HD and age (*p* = 0.0017, R −0.19)) that it was the young patients who spent more time on HD than others (Figure 16). The patients who died during the 6-month follow-up after vaccination for reasons other than COVID-19 were found to have had shorter weekly times on HD. Similarly, those with elevated pre-vaccination creatinine had higher post-vaccination AB levels. The creatinine level is known to be linked to age, sex, and nutritional status, mainly in terms of protein intake. In our study, the level of creatinine before HD correlated negatively with age (*p* = 1.62 × 10^−18^, R −0.59;). The patients who died during the 6-month follow-up after vaccination for reasons other than COVID-19 were found to have had lower creatinine concentrations (Table 3). No age differences were observed in the extreme terciles; however, in our analysis of patients who had no pre-vaccination history of SARS-CoV-2 infection, when we divided the group into two subgroups according to the median post-vaccination AB levels, it turned out these groups differed significantly as regards age (*p* = 0.04) (Table 5). Younger persons had higher AB levels. Both the entire group and the subgroup of patients with no history of infection were found to demonstrate a negative correlation between age and post-vaccination AB levels (*p* = 0.04 and 0 = 0.005, respectively) (Figure 11). Other authors have also confirmed that young patients show a better humoral response to vaccination [11,17,18].

Evidence that humoral immunity is dependent on cellular immunity is found in the fact that there was a positive correlation between post-vaccination AB levels and the post-vaccination cellular response rate (*p*= 8.59 × 10^–11^; R 0.43) (Figure 14). Melin, who studied a group of 50 patients in Sweden, arrived at similar outcomes [13].

### 4.2. Cellular Immunity

Positive cellular immunity developed in 72% of the patients. This result was lower than that obtained in the case of humoral immunity. The same results were obtained from Strengert et al. in their group of HD patients studied in Germany [12]. It is presently difficult to interpret the outcome of their paper as there are no clinical studies available on how cellular immunity protects the system against SARS-CoV-2 infection [19,20]. As mentioned earlier, cellular immunity correlates positively with humoral immunity, which is why it appears that both forms of response can be considered in parallel. Other researchers have found a similar relationship [12,21]. A worse cellular response, when compared to the humoral response, may be linked to chronic inflammation found in IHD patients, which leads to T cell exhaustion and thus a reduced production of IFNγ [22,23]. Although this group show elevated cytokines, such as IFNγ, TNF-α, IL-8, and CCL-2, studies on the response of CD4+ and CD8+ T cells to mitogen found that it was significantly reduced [24]. In a group of patients on HD in Sweden, a positive cellular response was only observed in 58% of them [13]. Cellular immunity varied between studies: it was high at more than 90% in Sattler, Bertrand, and Stumpf, and low in Brosetta and Espi [14,25,26,27,28]. A number of researchers have associated this with the varied ages of the study subjects. In our positive and negative response subgroups, we were only able to observe differences with regard to pre- and post-vaccination anti-SARS-CoV-2 antibody levels and did not notice other differences in respect of the parameters studied. When determining the IFNγ level in response to the antigen, we observed a positive correlation with pre- and post-vaccination anti-SARS-CoV-2 antibody levels and a positive correlation with albumin concentrations (Figure 2, Figure 14, Figure 15, Figure 19 and Table 2). These relationships were found by Praet et al. for humoral response [29]. Papers on the humoral response to HBV vaccination, such as the meta-analysis by Ghamar-Chehreh, have implicated a significant impact on albumin concentrations on the resultant AB levels [30]. There are no studies available assessing the effects of albumin concentrations on cellular immunity. However, albumin level is known to influence the development of frailty syndrome, and to increase mortality due to cardiovascular causes and infections. Low albumin content is believed to result from malnutrition on the one hand and inflammation on the other [31]. One very important piece of information learned from the post-vaccination 6-month follow-up is that those patients who died during that time for reasons not related to COVID-19 had a worse cellular response at 1 month after vaccination. A good cellular response appears to be related to a good prognosis for survival. Low albumin patients were among those who contracted COVID-19 during the 6-month follow-up.

### 4.3. Humoral Immunity and the Type of Vaccine

The AB levels at 1 month after the second dose were higher in the Moderna (mRNA-1273) group than in the Comirnaty (BNT162b2) group. The small size of the Moderna group (12 vs. 269) makes this result unreliable. A large study assessing the humoral and cellular response in dialysis patients in the USA (543 persons) concluded that those inoculated with the mRNA-1273 vaccine had mean responses that were mainly larger than the responses in the BNT162b2 vaccine recipients and were significantly more likely to achieve higher antibody thresholds thought to be required for preventing infection. This is associated with the different antigen doses contained in the vaccines. The BNT162b2 and the mRNA-1273 vaccines contain 30 µg and 100 µg of the viral antigen, respectively [15].

### 4.4. Neutrophil-to-Lymphocyte Ratio and Neutrophil Count

The group with no pre-vaccination history of SARS-CoV-2 infection was divided into terciles depending on their AB levels at 1 month after the 2nd dose of the vaccine: group 1 (<237 AU/mL), group 2 (237–970 AU/mL), and group 3 (>970 AU/mL). Between the extreme groups (group 1 versus 3), group 3 was found to have a higher cellular response, lower neutrophil concentrations, and lower neutrophil-to-lymphocyte (NLR) ratio values (Figure 8, Figure 9 and Figure 10). In addition, a negative correlation was found between AB levels at 1 month after inoculation with a SARS-CoV-2 vaccine and the NLR. The NLR has previously been reported as an inflammatory marker in end-stage renal disease (ESRD) patients. One study demonstrated that the NLR positively correlated with tumor necrosis factor-α (TNF-α) in 61 ESRD patients receiving PD or HD for ≥6 months [32]. The study showed that the NLR was significantly higher in PD patients than HD patients. A recent study enrolled 100 ESRD patients on maintenance HD for ≥3 months [32]. The study revealed that the correlation relationships between the NLR and hs-CRP were statistically robust. In the study conducted by Li et al. on 611 non-dialysis patients with ESRD in a cross-sectional study, the NLR was positively correlated with hs-CRP in non-dialysis ESRD patients, which was in line with other studies on PD or HD patients [32,33,34]. The NLR has proven to be a predictor of survival in HD patients [34,35]. Retrospectively, Catabay et al. compared the mortality predictability of the NLR among 108,548 incident HD patients [35]. The authors found a high NLR in incident HD patients predicted mortality, especially in the short term. In another study, Ouellet et al. evaluated the NLR as a predictor of survival in 5782 incident HD patients [36]. The NLR is superior to the total WBC count for the prediction of all-cause mortality in incident and prevalent HD patients and was identified as a novel and robust predictor of all-cause mortality in those patients [34]. In our study group, the patients who died during the 6-month follow-up after vaccination for reasons other than COVID-19 were found to have had a lower NLR, as well (Table 3). This parameter is a marker for severe COVID-19 in dialysis patients [37,38]. A higher NLR in HD patients can be considered to be associated with more pronounced inflammation and a higher mortality rate. There are currently no publications available assessing the magnitude of humoral response to vaccination depending on the NLR. The results obtained in the present study may be indicative of which group may require vaccination with a higher antigen dose, for instance, by choosing the Moderna vaccine over the Pfizer-BioNTech jab.

In both the entire study group and the subgroup of patients with no pre-vaccination history of SARS-CoV-2 infection, a higher neutrophil count was associated with a poorer humoral response to vaccination (Figure 3, Figure 4, Figure 5, Figure 6 and Figure 7). This may be related to the chronic inflammation caused by co-morbidities on the one hand, and hemodialysis itself on the other. Dialysis has been proven to increase the count and activation of neutrophils. Patients with an elevated neutrophil count most probably have a more exacerbated chronic inflammation. This is very likely to reduce the humoral response to vaccination [38]. There is no data available on evaluating HD patients’ response to vaccination according to their neutrophil count. This relationship was confirmed by the negative correlation between the neutrophil count and the AB level at 1 month after the 2nd dose of SARS-CoV-2 vaccine both in the entire group and in the subgroup of patients with no pre-vaccination history of COVID-19 (*p* = 0.03, R −0.13; and *p* = 0.003, R −0.25, respectively) (Figure 17). In the group of patients who died during the 6-month follow-up after vaccination for reasons other than COVID-19, significantly higher neutrophil counts were found; possibly, this parameter may become a marker for mortality in HD patients, alongside the NLR (Table 3).

### 4.5. Hemoglobin Level

In the group of patients with no pre-vaccination history of SARS-CoV-2 infection, post-vaccination humoral immunity correlated positively with hemoglobin levels. As mentioned above, humoral immunity was dependent on the markers for chronic inflammation and malnutrition. On hemodialysis, abnormal hemoglobin is associated with erythropoietin resistance, which is known to be a state predominantly dependent on inflammation. The available literature does not offer data on humoral immunity, erythropoietin resistance, or the HGB level. The patients who contracted COVID-19 during the post-vaccination 6-month follow-up had significantly lower HGB levels.

## 5. Summary

Patients on hemodialysis with stage 5 chronic kidney disease according to the KDIGO classification are a group characterized by a considerable SARS-CoV-2 vaccine-induced seroconversion rate. The strongest factors influencing the AB level after vaccination are a pre-vaccination history of SARS-CoV-2 infection, age, the NLR, neutrophil absolute count, and the hemoglobin level. Cellular immunity was higher in patients with a pre-vaccination history of SARS-CoV-2 infection. The albumin level affected the volume of cellular immunity. During the 6-month follow-up, 12 patients died for reasons not attributable to COVID-19. Those who died within 6-months of being vaccinated had lower cellular immunity at 1 month after the second dose. Cellular immunity and humoral immunity were positively mutually dependent.

High levels of albumin and hemoglobin, low neutrophil count, and a reduced NLR translated to a better response to vaccination. All these factors change with exacerbating inflammation and malnutrition, which are likely to affect the efficacy of the vaccination.

Probably the vaccine of choice for patients with low albumin and hemoglobin levels, a high NLR, and a high neutrophil count should be the mRNA-1273 vaccine, which contains three times the dose of the viral mRNA than the other product.

Strengths: Our study was conducted on a large group of patients from a number of dialysis centers. The study group included both COVID-19 convalescents and patients with no history of the infection, which allowed us to run both joint and separate analyses. The post-vaccination 6-month follow-up enabled an evaluation of the incidence of COVID-19 and mortality in the study group.

Limitations: Our study was not designed or powered to assess the effects of vaccination on the incidence of COVID-19; our follow-up lasted 6 months after the second dose of vaccination. A larger population and a longer follow-up, including statistical modeling to account for the dynamics of the epidemic and the effects of the virus variants, will be required to evaluate the effects of vaccination on COVID-19-related hospitalization rates and mortality in hemodialysis patients.

## 6. Conclusions

The SARS-CoV-2 vaccine-induced seroconversion rate is high in hemodialyzed patients. The factors influencing the AB level after vaccination are a pre-vaccination history of SARS-CoV-2 infection, age, the NLR, neutrophil absolute count, and the hemoglobin level. The factors influencing cellular immunity include a pre-vaccination history of SARS-CoV-2 infection and the albumin level. A positive cellular response to vaccination is a positive factor in reducing all-cause mortality. Cellular immunity and humoral immunity are positively mutually dependent.

## Figures and Tables

**Figure 1 biomedicines-10-00636-f001:**
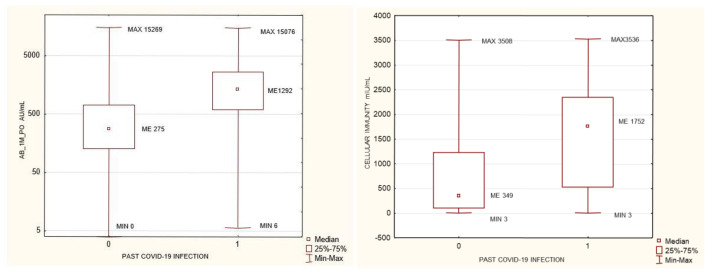
Anti-SARS-CoV-2 antibody levels prior to and after vaccination in groups that developed positive or negative cellular immunity. *p* = 0.004 and *p* = 1.77 × 10^–05^, respectively. Mann–Whitney U test. AB_1M_PO- anti SARS-CoV2 antibody level one month after completed vaccination, CELLULAR IMMUNITY after completed vaccination: INFγ concentration > 200 mIU/mL positive, <100 mIU/mL negative.

**Figure 2 biomedicines-10-00636-f002:**
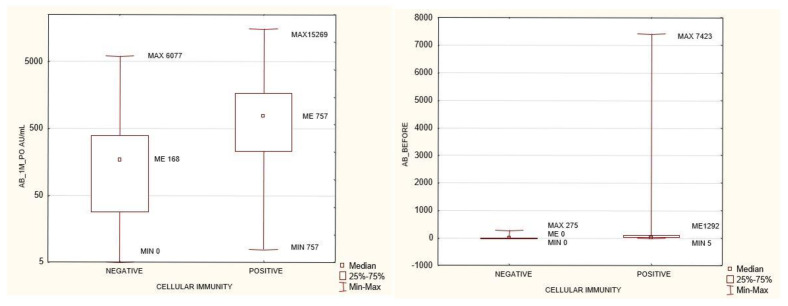
Anti-SARS-CoV-2 antibody levels prior to and after vaccination in groups that developed positive or negative cellular immunity. P = 0.004 and *p* = 1.77 × 10^–05^, respectively. Mann-Whitney U test. AB_1M_PO- anti SARS-CoV2 antibody level one month after completed vaccination, CELLULAR IMMUNITY after completed vaccination: INFγ concentration >200 mIU/mL positive, <100 mIU/mL negative.

**Figure 3 biomedicines-10-00636-f003:**
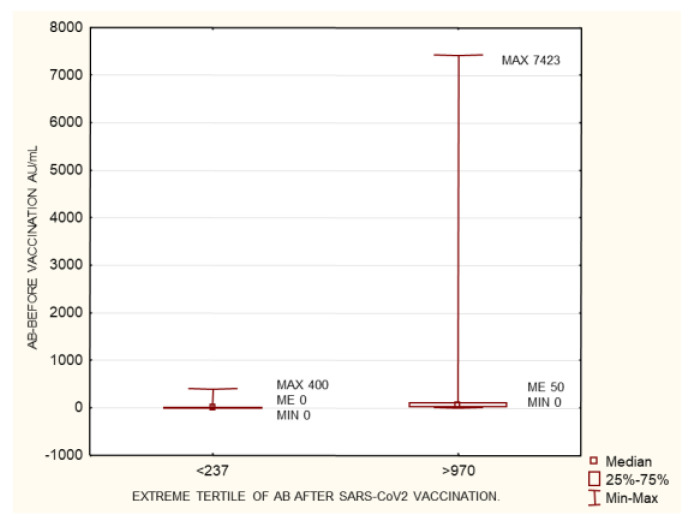
Statistically significant differences in antibody levels against SARS-CoV2 before vaccination between the extreme terciles of AB levels after SARS-CoV-2 vaccination <237 AU/mL and >970 AU/mL *p* = 5.26 × 10^–11^. Mann–Whitney U test. AB: antibody.

**Figure 4 biomedicines-10-00636-f004:**
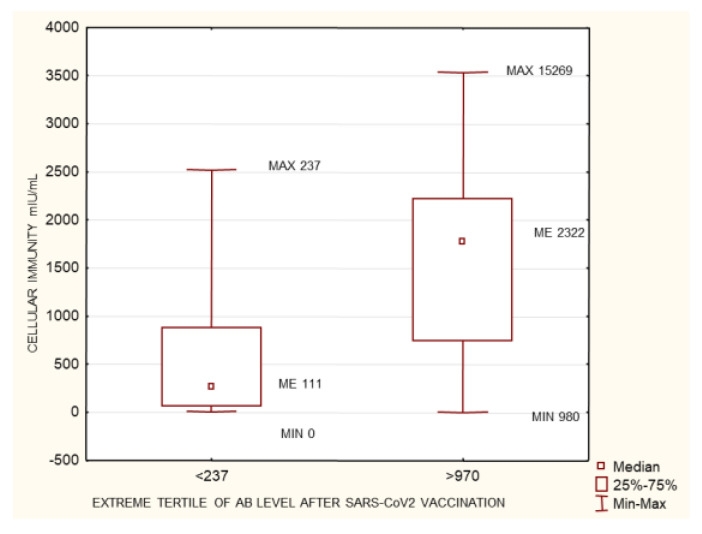
Statistically significant differences in cellular immunity between the extreme terciles of AB levels after SARS-CoV-2 vaccination <237 AU/mL and >970 AU/mL *p* = 5.66 × 10^–08^. Mann–Whitney U test. CELLULAR IMMUNITY after completed vaccination: INFγ concentration mIU/mL, AB: antibody.

**Figure 5 biomedicines-10-00636-f005:**
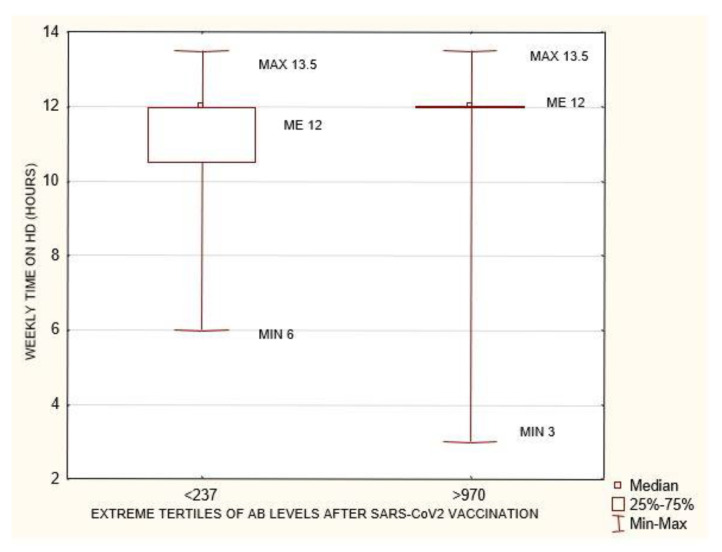
Statistically significant differences in weekly time on HD between the extreme terciles of AB levels after SARS-CoV-2 vaccination <237 AU/mL and >970 AU/mL *p* = 0.018. Mann–Whitney U test. AB: antibody.

**Figure 6 biomedicines-10-00636-f006:**
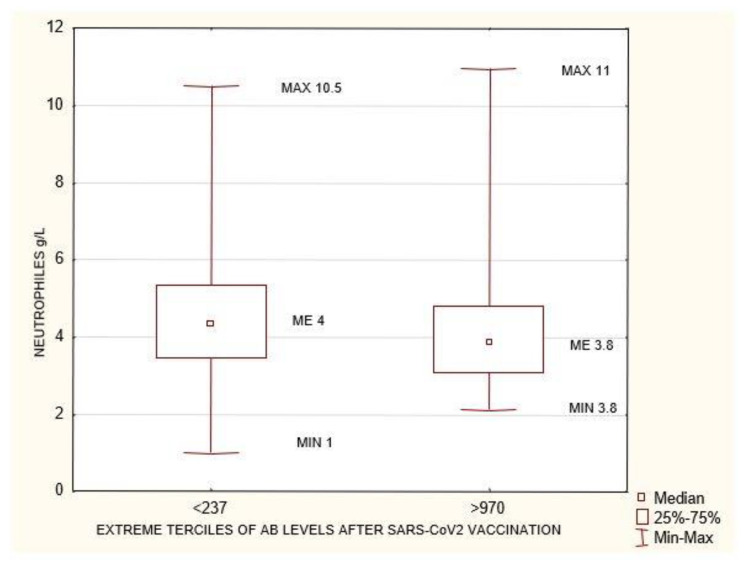
Statistically significant differences in neutrophiles concentration (g/L) between the extreme terciles of AB levels after SARS-CoV-2 vaccination <237 AU/mL and >970 AU/mL *p* = 0.03. Mann–Whitney U test. AB: antibody.

**Figure 7 biomedicines-10-00636-f007:**
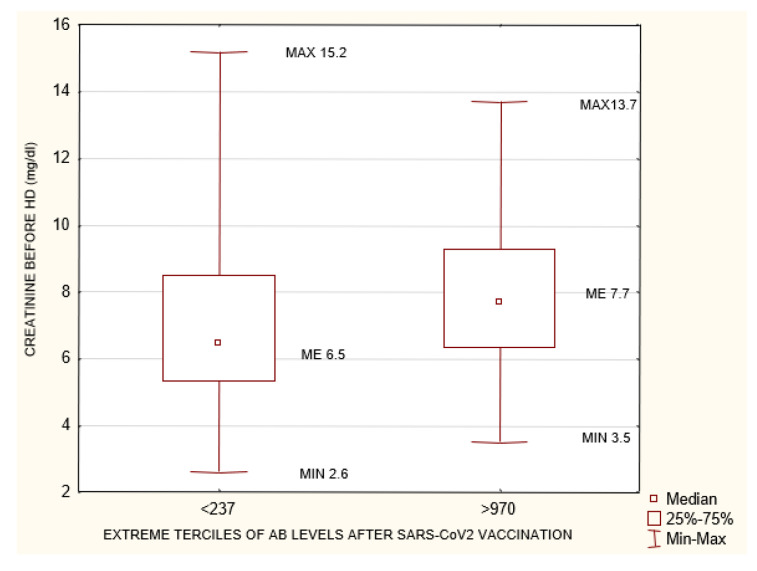
Statistically significant differences in creatinine concentration (mg/dL) before HD between the extreme terciles of AB levels after SARS-CoV-2 vaccination <237 AU/mL and >970 AU/mL *p* = 0.03. Mann–Whitney U test. AB: antibody.

**Figure 8 biomedicines-10-00636-f008:**
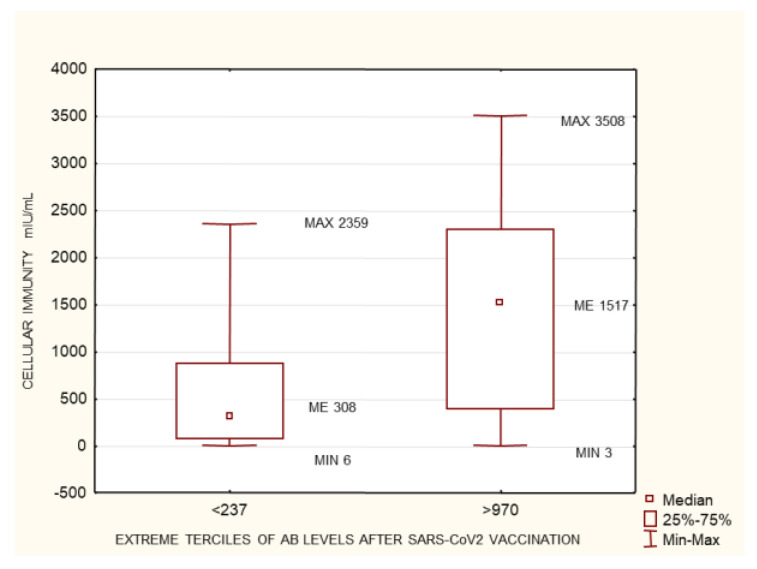
The group of patients with no history of SARS-CoV-2 infection prior to vaccination. Statistically significant differences in cellular immunity between the extreme terciles of post-SARS-CoV-2 vaccination antibody levels <237 AU/mL and >970 AU/mL. *p* = 1.14 × 10^–11^. Mann–Whitney U test. AB: antibody.

**Figure 9 biomedicines-10-00636-f009:**
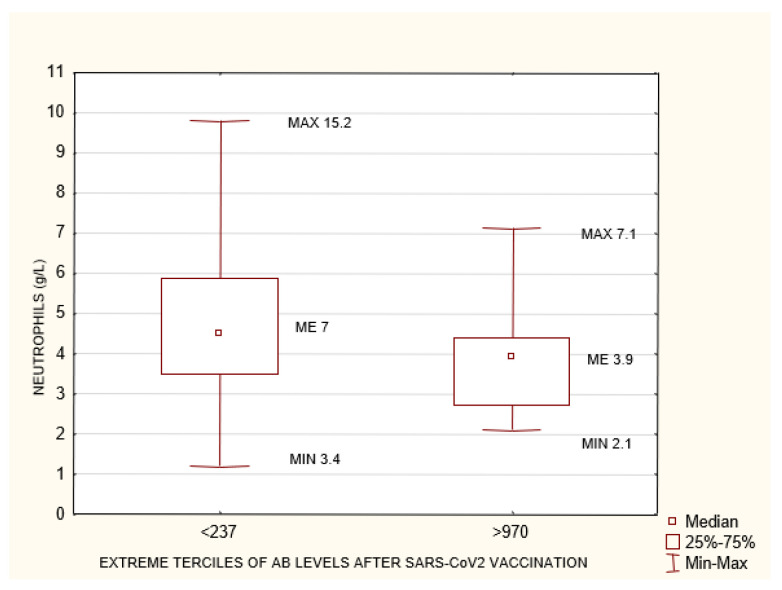
The group of patients with no history of SARS-CoV-2 infection prior to vaccination. Statistically significant differences in neutrophiles concentration (g/L) between the extreme terciles of post-SARS-CoV-2 vaccination antibody levels <237 AU/mL and >970 AU/mL. Mann–Whitney U test. *p* = 0.03. AB: antibody.

**Figure 10 biomedicines-10-00636-f010:**
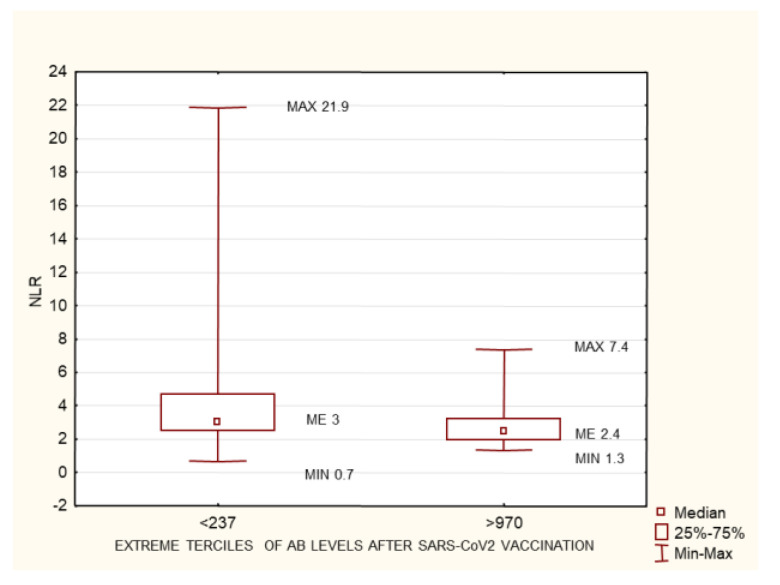
The group of patients with no history of SARS-CoV-2 infection prior to vaccination. Statistically significant differences in the neutrophil-to-lymphocyte ratio between the extreme terciles of post-SARS-CoV-2 vaccination antibody levels <237 AU/mL and >970 AU/mL. Mann–Whitney U test. *p* = 0.03. AB: antibody, NLR: neutrophil-to-lymphocyte ratio.

**Figure 11 biomedicines-10-00636-f011:**
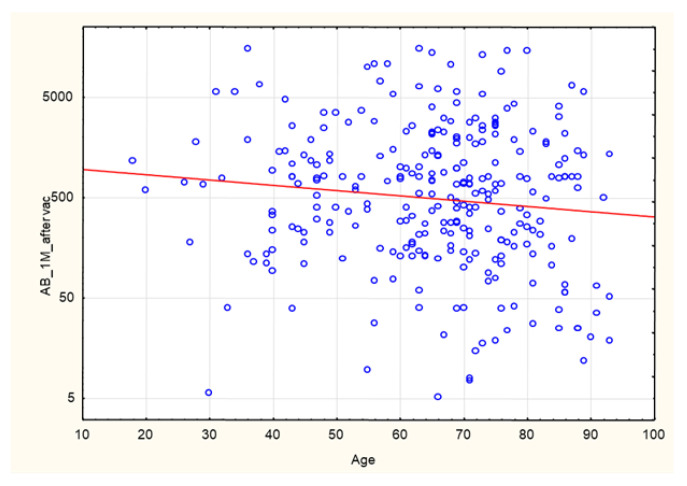
Negative correlation between age (years) and anti-SARS-CoV-2 antibody levels (AU/mL) at 1 month after completion of the vaccination course (*p* = 0.04; R −0.12). Spearman’s rank correlation test. AB: antibody.

**Figure 12 biomedicines-10-00636-f012:**
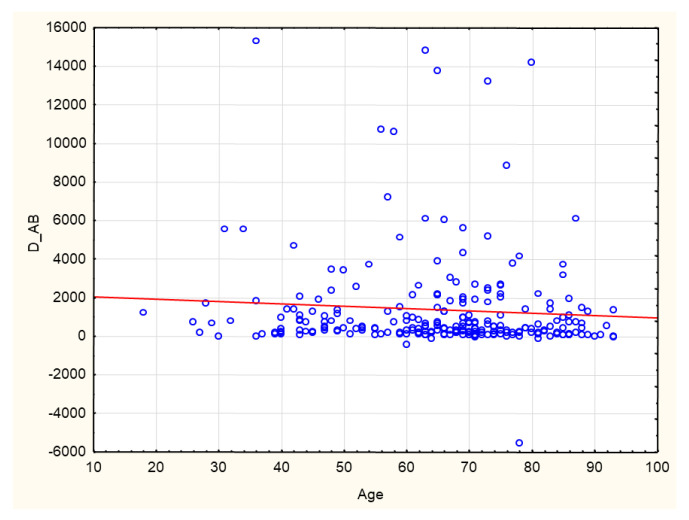
Negative correlation between the delta of antibodies (AU/mL) before and after vaccination and age (years). (*p* = 0.026; R −0.14) Spearman’s rank correlation test. D_AB: delta of antibody. Antibody levels before and after completion of the vaccination course.

**Figure 13 biomedicines-10-00636-f013:**
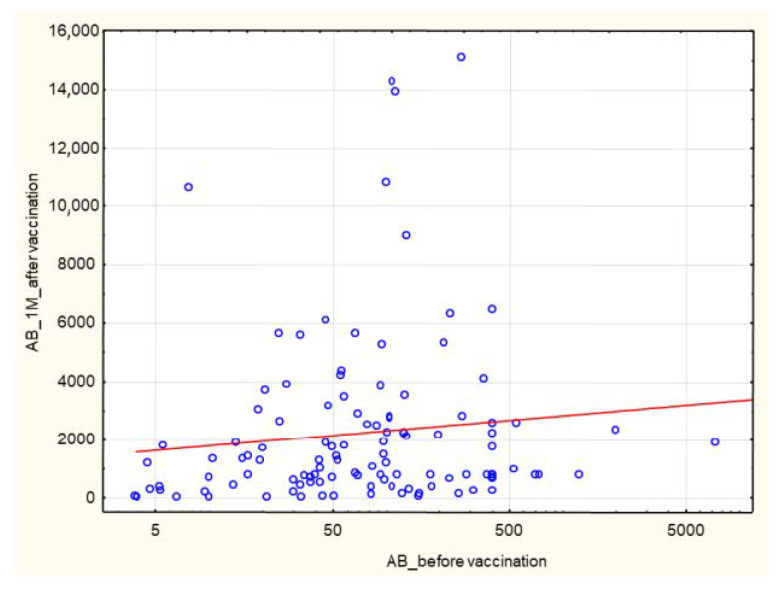
Positive correlation between pre-vaccination AB levels and the AB levels at 1 month after the 2nd dose of SARS-CoV-2 vaccine (AU/mL). (*p* = 1.1 × 10^–14^, R 0.47). Spearman’s rank correlation test. AB: antibody.

**Figure 14 biomedicines-10-00636-f014:**
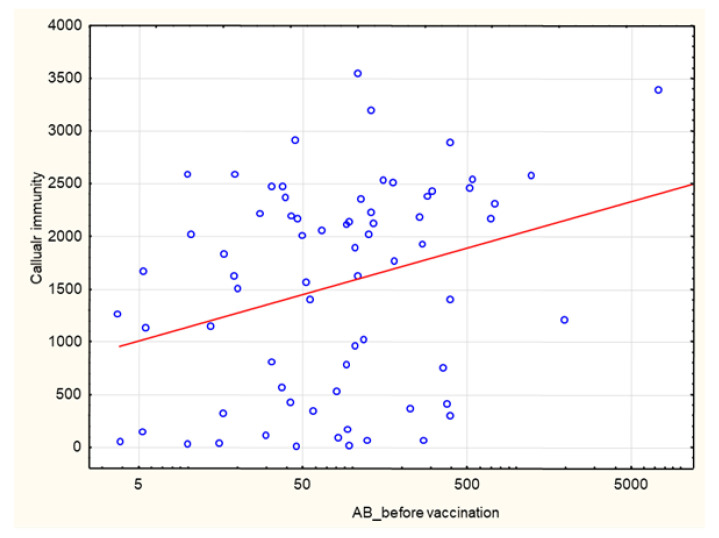
Positive correlation between pre-vaccination AB levels (AU/mL) and cellular immunity (INFγ concentration mIU/mL) at 1 month after the 2nd dose of SARS-CoV-2 vaccine (*p* = 7.95 × 10^–^^09^, R 0.43). Spearman’s rank correlation test.

**Figure 15 biomedicines-10-00636-f015:**
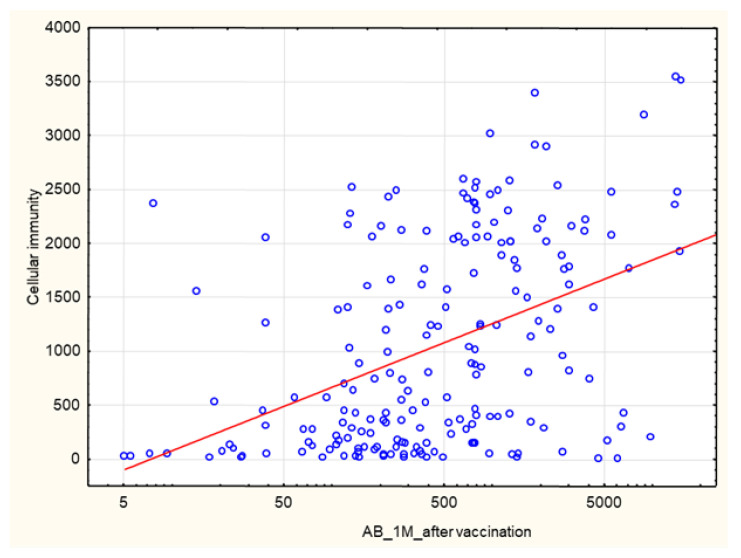
Positive correlation between AB levels (AU/mL) at 1 month after the 2nd dose of SARS-CoV-2 vaccine and cellular immunity (INFγ concentration mIU/mL) at the same time (*p* = 8.6 × 10^−11^, R 0.44). Spearman’s rank correlation test.

**Figure 16 biomedicines-10-00636-f016:**
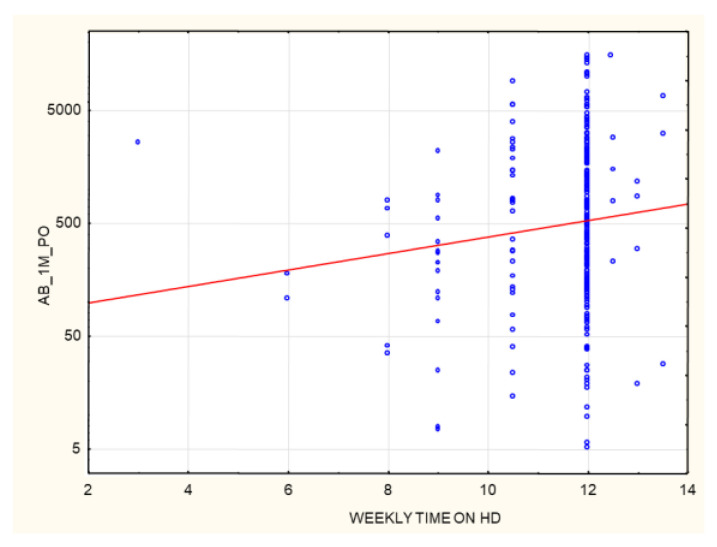
Positive correlation between AB levels (AU/mL) at 1 month after the 2nd dose of SARS-CoV-2 vaccine and the weekly time (hours) on HD (*p* = 0.017, R 0.14). Spearman’s rank correlation test. AB: antibody, HD: hemodialysis.

**Figure 17 biomedicines-10-00636-f017:**
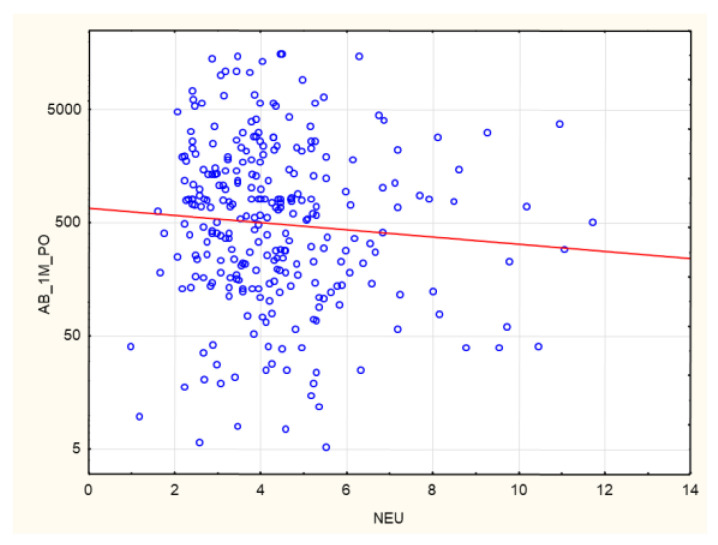
Negative correlation between AB levels (AU/mL) at 1 month after the 2nd dose of SARS-CoV-2 vaccine and neutrophil concentrations (g/L) (*p* = 0.03, R −0.13). Spearman’s rank correlation test. AB: antibody, NEU: neutrophil.

**Figure 18 biomedicines-10-00636-f018:**
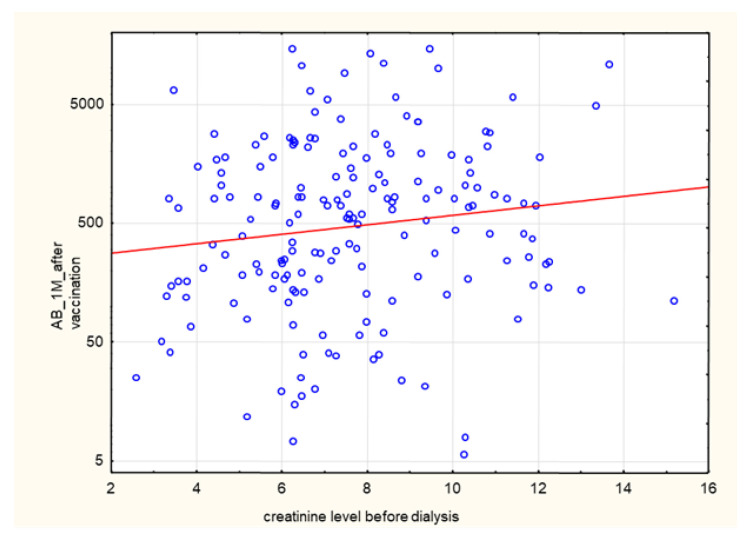
Positive correlation between AB levels (AU/mL) at 1 month after the 2nd dose of SARS-CoV-2 vaccine and creatinine concentrations before HD (*p* = 0.018, R 0.17). Spearman’s rank correlation test. AB: antibody.

**Figure 19 biomedicines-10-00636-f019:**
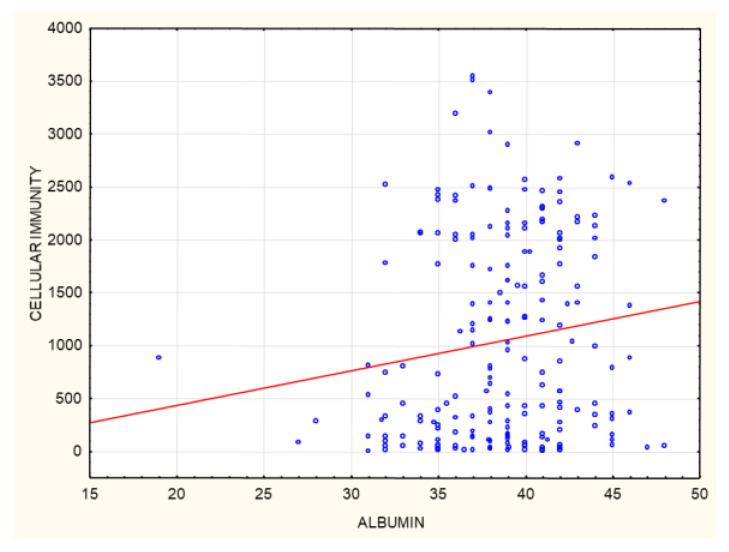
Positive correlation between cellular immunity (INFγ concentration mIU/mL) at 1 month after the 2nd dose of SARS-CoV-2 vaccine and albumin concentrations (g/L) (*p* = 0.02, R 0.16). Spearman’s rank correlation test.

**Table 1 biomedicines-10-00636-t001:** Characteristics of the study group.

	N	Median	Min	Max
AGE (years)	281	68	18	93
AB before vaccination AU/mL	281	0	0	7423
AB 1-month post-vaccination AU/mL	281	576.92	0	15,269.23
Cell immunity 1-month post-vaccination mIU/mL	202	757.05	3.032	3535.69
Time on hemodialysis (months)	281	38	1	411
KT/V	281	1.38	0.67	3.48
Time of dialysis per week (hours)	281	12	3	13.5
BMI (kg/m^2^)	281	27	15.2	45.5
SGA-DMS	90	12	8	32
Albumin (g/L)	281	39	19	48
Transferrin (g/L)	159	1.64	0.52	2.91
Hgb (g/dL)	281	10.8	6	15.1
PTH Pg/mL)	281	6.54	0.33	1148
Ferritin (ng/mL)	281	109	1	1367.09
Lymphocytes (g/L)	281	1.43	0.2	3.5
Neutrophiles (g/L)	281	4.02	1	11.75
Urea before HD (mg/dL)	281	111.8	53	256
Creatinine before HD (mg/dL)	180	7.32	2.6	15.21
Anti HBS antibody (mIU/mL)	268	48.15	0	1000
Delta of antibodies (AU/mL)	281	450.46	−5549.92	15,269.23

SGA-DMS subjective global assessment-dialysis malnutrition scoring, AB: antibody, Kt/V: measurement of the efficacy of a hemodialysis session, BMI: body mass index, HGB: hemoglobin, PTH: parathyroid hormone, HD: hemodialysis, HBS hepatitis B surface antigen.

**Table 2 biomedicines-10-00636-t002:** Anti-SARS-CoV-2 antibody levels prior to and after vaccination in groups that developed positive or borderline cellular immunity.

	Cell. Immunity Borderline Interferon Gamma 100–200 mIU/mL N = 38			Cell. Immunity Positive Interferon Gamma >200 mIU/mL N = 146			
AB-Antibody	Me	Min	Max	Me	Min	Max	*p*
AB before vac. AU/mL	0	0	95	1292.3	5.4	7423	0.001
AB after vac. AU/mL	255.53	23.4	5269.23	757.12	7.76	15,269.23	0.0037

**Table 3 biomedicines-10-00636-t003:** Statistically significant differences between the groups with and without death during the 6-month follow-up after the end of the vaccination course.

	Death during the 6-Month Follow-Up *n* = 12			No Death during the 6-Month Follow-Up *n* = 269			
	Me	Min	Max	Me	Min	Max	*p*
Cell Immunity 1-month post-vac. mIU/mL	158.17	4.09	775.57	805.39	3.03	3535.69	0.007
Albumin (g/L)	36.6	32	42	39	19	48	0.019
Neutrophils (g/L)	4.785	3.06	8.63	4	1	11.75	0.04
NLR	12.88	3.44	36	2.92	0.68	10.33	0.019
Creatinine before HD (mg/dL)	5.21	2.6	9.7	7.45	3.2	15.21	0.007
Time on HD per week (hours)	10.5	8	12	12	3	13.5	0.019

NLR: neutrophil-to-lymphocyte ratio, HD hemodialysis.

**Table 4 biomedicines-10-00636-t004:** Statistically significant differences between the groups with and without SARS-CoV-2 infection during the 6-month follow-up after the end of the vaccination course.

	**COVID-19 during the 6-Month Follow-Up after Vaccination *n* = 8**			**No COVID-19 during the 6-Month Follow-Up after Vaccination *n* = 273**			
	**Me**	**Min**	**Max**	**Me**	**Min**	**Max**	** *p* **
Hgb (g/dL)	9.25	6	12.2	10.854	7.29	15.1	0.02
Albumin (g/L)	34.5	19	39.8	39	27	48	0.002

HGB hemoglobin level.

**Table 5 biomedicines-10-00636-t005:** Statistically significant differences within the group of patients who had no SARS-CoV-2 infection prior to vaccination. They were divided into patients below and those above or equal the median AB level following vaccination ME 570 AU/mL.

	Group 1 <570 AU/mL			Group 2 ≥570 AU/mL			
	Me	Min	Max	Me	Min	Max	*p*
Cell. Immunity 1-month post-vac. mIU/mL	264.0	5.7	2359.1	62	26	88	0.0017
Age (years)	68	27	92	62	26	88	0.04

## Data Availability

Data supporting reported results can be found with the corresponding author.

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
