# Peer review of "SARS-CoV-2 mRNA Vaccine-Induced Cellular and Humoral Immunity in Hemodialysis Patients"

_biomedicines, 2022, doi:10.3390/biomedicines10030636_

Round 1

Reviewer 1 Report

I woudl suggest the authors to take informations also from this articles because they are very interesting

DOI 10.3390/microorganisms9040793

DOI 10.2147/RMHP.S284557

Author Response

Dear Reviewers and Editor

Thank you very much for critical review of our article “ SARS-CoV-2 mRNA vaccine-induced cellular and humoral immunity in hemodialysis patients.”. We have carefully considered your suggestions and have revised our manuscript accordingly; we hope that these changes meet with your approval.

Ewa Kwiatkowska

Reviewer 2 Report

Please change the European formatting in Table 1 (123,00) to conventional decimal places per journal convention (123.00).

The data presentation has been significantly improved. 

Author Response

(The authors gave the same response as above.)

Reviewer 3 Report

General: This is a very nicely written article, with only minor errors. Also, it touches an important aspect of COVID-19 vaccines and their effects. Overall, the story flow is good, and it was a thoroughly conducted study. However, I think a sampling between Dose-1 and Dose-2 would have made the study much stronger and the authors could then, put their claims in a much assertive manner (See the relevant comment in Specific comments). Although I am aware that this would a significant amount of labor and financial logistics to the study!

Also, I would insist that the ethics statement to be moved from Line 161 to around 129-130. It should be clearly visible in the front end of the methodology.

I would suggest the manuscript to be accepted with minor corrections. 

Specific:

Abstract: Please make sure that the results are written in past tense and not in present tense. Also, please use the full names for abbreviations such as CKD, IHD etc. on first usage. If abstract becomes too large due to the full forms, maybe consider putting a different section, listing the abbreviations (similar to how they do in book chapters or PhD theses).

Table 1: Please put the contents of Column 1 in 'Sentence case' and not in 'All Capital' case. Same for other tables.

Line 169: Put these in Sentence case and not in All Capitals.

Line 179: Indicate the product details for products such as 'LIAISON assay by DiaSorin' in a 'Product (Product code, Manufacturer, City, Country)'format.  For example, the citation for this example would be 'LIAISON assay (DiaSorin S.p.A. Via Crescentino, Saluggia (VC), Italy)

Line 371: Please reverse the sequence of  (R 0.47; p=1.1E-14). Same for other places such as Line 402-442. 

Line 518: (8.59915E-11; R 0.43) should be (p = 8.59915E-11; R 0.43). Also, my suggestion would be keep the p values to 2 decimal places if they are very small and indicated as 'e.g. 8.59E-11 instead of 8.59915E-11'.

Line 641-643: This is a very strong claim, but the authors have not conducted post Dose-1 sampling which would back the claim in a much  stronger way than it is doing now. Also, the authors have not sampled at multiple time points after Dose-2 which would have helped to build a very strong case to back this claim. I would ask the authors to either back their claim with very strong study citations (in the lack of their own multiple point data) or, tone down these lines a bit. 

References: Please ensure that the titles of all the references are in 'Sentence Case'.

Author Response

Dear Reviewers and Editor

Thank you very much for critical review of our article “ SARS-CoV-2 mRNA vaccine-induced cellular and humoral immunity in hemodialysis patients.”. We have carefully considered your suggestions and have revised our manuscript accordingly; we hope that these changes meet with your approval.

Ewa Kwiatkowska

This manuscript is a resubmission of an earlier submission. The following is a list of the peer review reports and author responses from that submission.

Round 1

Reviewer 1 Report

General Comments: This is a great paper and is well written, even if its somewhat descriptive. I congratulate the authors for producing a good paper. There are very few, minor corrections that are needed before the manuscript is publishable. I suggest minor corrections (specified below) before accept.

Specific Comments:

Line 98: Please correct  IU/ml to  IU/mL throughout the manuscript

Line 145: In Table 1, please elaborate what individual short forms such as ST, BMI etc. mean. Similarly, does AB (row 1 and 2) refer to antibodies (last 2 rows)? As there is no elaboration on this, I am confused between these rows and last 2 rows. Same for Table 2

Author Response

Dear Reviewers,

Thank you very much for critical review of our article “ SARS-CoV-2 mRNA vaccine-induced cellular and humoral immunity in haemodialysis patients.”. We have carefully considered yours suggestions and have revised our manuscript accordingly; we hope that these changes meet with your approval. The manuscript has been proof-read by a certified English language translator, who specializes in medical English.

Ewa Kwiatkowska

Reviewer 2 Report

Some careless editing -

Line 96: HBV surface antigen on a 0-,1-, and - month schedule)

Some issues with the methodology -

Lines 130 - 135: the multitude of factors analysed makes this look like an exploratory study; personally I think the tabulation (Table 1) of those characteristics is not particularly relevant

Line 154: for "unlimited" high range, the authors should remain cautious regarding such limitations as the hook effect 

Tables 2 - 7 are presented in a manner that is very difficult to read. Min/Max, Mean / Median and Standard Deviation values could be better presented. Perhaps boxplots?

The correlations are statistically significant but look to me rather weak (R = -0.14 or +0.44 do not seem particularly strong to me). 

Author Response

(The authors gave the same response as above.)

Reviewer 3 Report

For the first paragraph of the introduction please provide the bibliography.

Please mention also the inclusion and exclusion criteria of the patients in the study.

I would suggest the authors to provide a more concise conclusion and move that part to the discussion section.

Author Response

(The authors gave the same response as above.)

Round 2

Reviewer 2 Report

Thank you for considering my comments.

Line 304: R = 0.47 is usually considered "moderate" and not very strong

https://sphweb.bumc.bu.edu/otlt/MPH-Modules/PH717-QuantCore/PH717-Module9-Correlation-Regression/PH717-Module9-Correlation-Regression4.html

Line 527: Anaemia is an important problem in patients with ESRF. Patients with haemoglobinopathy present an even more complicated situation. Here the meaning of "abnormal hemoglobin" is not clear - does it mean abnormal Hb level?

Line 539: Cellular immunity... albumin level - remember this study focuses on association, and it would be too early to state that certain causative mechanisms are confirmed in this study. 

Line 540-541: meaning is unclear

Suggest language editing and re-organization of the article to make the flow more coherent. 

Round 3

Reviewer 2 Report

I hope the information presented in this paper can inform COVID-19 research community. The significance of the study, at the end of the day, is for the community to judge after publication. 

Please suggest authors to use language editing service . The current language quality is still unsatisfactory.
